# Non-Hermitian Dirac cones with valley-dependent lifetimes

Xinrong Xie ®[1,2,3,4,8], Fei Ma[1,2,3,4,8], W. B. Rui[5,6], Zhaozhen Dong[1,2,3,4], Yulin Du[1,2,3,4], Wentao Xie[7], Y. X. Zhao ®[5,6], Hongsheng Chen ®[1,2,3,4], Fei Gao ®[1,2,3,4] ✉ & Haoran Xue ®[7] ✉

Relativistic quasiparticles emerging from band degeneracies in crystals play crucial roles in the transport and topological properties of materials and metamaterials. Quasiparticles are commonly described by Hermitian Hamiltonians, with non-Hermiticity usually considered detrimental. In this work, we show that such an assumption of Hermiticity can be lifted to bring quasiparticles into non-Hermitian regime. We propose a concrete lattice model containing two Dirac cones with valley-dependent lifetimes. The lifetime contrast enables an ultra-strong valley selection rule: only one valley can survive in the long-time limit regardless of the excitation, lattice shape and other details. This property leads to an effective parity anomaly with a single Dirac cone and offers a simple way to generate vortex states. Additionally, extending non-Hermitian features to boundaries generates valley kink states with valley-locked lifetimes, making them effectively unidirectional and more resistant against inter-valley scattering. All these phenomena are experimentally demonstrated in a non-Hermitian electric circuit lattice.

Dirac equation, proposed by Paul Dirac in 1928 as the first reconciliation of special relativity and quantum mechanics, has a profound impact on the development of many aspects of modern physics[1]. While the Dirac equation was mainly studied in the context of particle physics and quantum field theories in the early years, people later found that Dirac physics can also be accessed in crystals with band degeneracies called Dirac points[2]. In the low-energy limit, the states around a Dirac point are effectively governed by the Dirac equation and thus behave as Dirac quasiparticles[3]. Beyond Dirac, other types of quasiparticles can also emerge in certain band degeneracies[4,5], such as Weyl quasiparticles[6] and even those not allowed in particle physics[7]. These quasiparticles, apart from their own theoretical interest as fundamental particles, are responsible for many topological phenomena and offer new insights for

controlling the transport in crystals. For example, Dirac quasiparticles in graphene give rise to the quantum Hall effect under a magnetic field[2,3], and their valley contrasting physics opens a route to manipulate electrons and forms the basis for valleytronics[8,9].

In recent years, nonconservative systems have appeared as a new playground for studying band degeneracies[10,11]. In contrast to closed systems, nonconservative systems modeled by non-Hermitian Hamiltonians typically host exceptional points where both eigenvalues and eigenstates coalesce[12–14]. Dirac points and other Hermitian band degeneracies, on the other hand, are usually unstable under non-Hermitian perturbations[15–17] and can only survive under symmetry protection[18,19]. In the latter case, the Dirac quasiparticles still obey the conventional Hermitian Dirac equation and behave much like those in

[1]State Key Laboratory of Extreme Photonics and Instrumentation, International Joint Innovation Center, The Electromagnetics Academy at Zhejiang University, Zhejiang University, Haining, China. [2]ZJU-Hangzhou Global Scientific and Technological Innovation Center, Zhejiang University, Hangzhou, China. [3]Key Lab. of Advanced Micro/Nano Electronic Devices & Smart Systems of Zhejiang, Jinhua Institute of Zhejiang University, Zhejiang University, Jinhua, China. [4]Shaoxing Institute of Zhejiang University, Zhejiang University, Shaoxing, China. [5]Department of Physics and HKU-UCAS Joint Institute for Theoretical and Computational Physics at Hong Kong, The University of Hong Kong, Pokfulam Road, Hong Kong, China. [6]HK Institute of Quantum Science & Technology, The University of Hong Kong, Pokfulam Road, Hong Kong, China. [7]Department of Physics, The Chinese University of Hong Kong, Shatin, Hong Kong SAR, China. [8]These authors contributed equally: Xinrong Xie, Fei Ma. ✉e-mail: gaofeizju@zju.edu.cn; haoranxue@cuhk.edu.hk

Hermitian systems[18,19]. To date, it is still unclear whether unique non-Hermitian physics can be obtained for conventional band degeneracies like Dirac points.

In this work, we design and experimentally realize a non-Hermitian extension of the graphene lattice with a complex conjugate pair of Dirac cones. Unlike all previously realized Dirac cones in either Hermitian or non-Hermitian systems[3,19], the Dirac cones in our study are intrinsically non-Hermitian. That is, the Dirac quasiparticles are governed by non-Hermitian Dirac Hamiltonians with a valley-dependent background imaginary term (see Fig. 1 and Eq. (2) below). This also differs from the trivial scenario where a uniform background loss added to a Hermitian graphene lattice induces the same imaginary term for both valleys. Given these differences, we dub the excitations around the non-Hermitian Dirac cones in our system non-Hermitian Dirac quasiparticles.

We fabricate a circuit lattice to implement the theoretical model. Owing to the sharp imaginary eigenvalue contrast, one of the Dirac cones with a shorter lifetime is wiped out in the long time limit, making our system effectively exhibit single Dirac cone physics. In the experiments, we consistently find that only one of the valleys can be excited without using any tailored excitation (Fig. 2). This situation also holds for the massive case where the Dirac cones are gapped by a staggered on-site potential, and vortex states can be simply excited by a point source (Fig. 3). We then use the massive non-Hermitian Dirac cones to construct mass-flipping interfaces. It is found that the valley kink states inherit the non-Hermitian properties from the bulk and have valley-dependent lifetimes. Their propagation is effectively unidirectional and the valley flipping is significantly suppressed due to the difference in the imaginary eigenvalues between the two valleys (Figs. 4 and 5). These results clearly reveal that Dirac cones can be induced in non-Hermitian lattices and can have interesting non-Hermitian physics beyond their Hermitian counterparts.

## Results

### Model for realizing non-Hermitian Dirac cones

Our starting point is the tight-binding model for graphene without spin-orbit coupling, as illustrated in the upper panel of Fig. 1a, which is known to host two Dirac cones at the K and K′ valleys (Fig. 1a, lower panel). The modes around the two valleys form the standard Dirac

quasiparticles, having real eigenvalues and orthogonal eigenstates and giving rise to various important physical phenomena[2], including Klein tunneling[20], quantum Hall effect[3], and valley contrasting physics[8]. Notably, when complex (but still Hermitian) next-nearest-neighbor (NNN) couplings that break the time-reversal symmetry (TRS) and on-site mass detuning are introduced, as proposed by Haldane in 1988[21], the two Dirac cones acquire different masses and exhibit unequal bandgaps with parity anomaly (Fig. 1b). In this case, only one Dirac cone can be accessed near zero energy and chiral edge states can be obtained when the Chern number is nonzero.

Based on the graphene model and the Haldane model, here we propose a non-Hermitian graphene model with real but nonreciprocal NNN couplings, as depicted in Fig. 1c. The corresponding Bloch Hamiltonian reads

$$H^{\mathrm{tb}}(\mathbf{k}) = \sum_{i=0}^{3} h_i^{\mathrm{tb}}(\mathbf{k})\sigma_i, \tag{1}$$

where $h_0^{\mathrm{tb}} = \sum_{i=1,2,3} 2[t_2 \cos(\mathbf{k}\cdot\mathbf{a}_i) - i\delta\sin(\mathbf{k}\cdot\mathbf{a}_i)]$, $h_1^{\mathrm{tb}} = t[1 + \cos(\mathbf{k}\cdot\mathbf{a}_2) + \cos(\mathbf{k}\cdot\mathbf{a}_3)]$, $h_2^{\mathrm{tb}} = t[\sin(\mathbf{k}\cdot\mathbf{a}_2) - \sin(\mathbf{k}\cdot\mathbf{a}_3)]$, $h_3^{\mathrm{tb}} = m$, $\sigma_0$ is the identity matrix, and $\sigma_{1,2,3}$ are the Pauli matrices. Here $\mathbf{k} = (k_x, k_y)$ is the wave-vector, $\mathbf{a}_1 = (1, 0)^{\mathrm{T}}$, $\mathbf{a}_2 = (-1/2, \sqrt{3}/2)^{\mathrm{T}}$ and $\mathbf{a}_3 = (-1/2, -\sqrt{3}/2)^{\mathrm{T}}$ are lattice vectors, $t$ is the nearest-neighbor coupling, $m$ is the on-site mass detuning and $t_2 \pm \delta$ is the non-Hermitian NNN coupling. We assume that all coupling terms are real, thus preserving the TRS.

A typical bulk dispersion of this lattice is plotted in Fig. 1d, where we see two Dirac-cone-like dispersions with opposite imaginary eigenvalues at the K and K′ valleys. Linearizing the lattice Hamiltonian around the valleys, we obtain the following valley-contrasting non-Hermitian Dirac Hamiltonians (see Supplementary Information for a comparison between the lattice Hamiltonian and the effective one):

$$H_{\mathrm{K/K'}}^{\mathrm{tb}}(\mathbf{q}) = (-3t_2 + \tau 3\sqrt{3}i\delta)\sigma_0 - \tau v q_x \sigma_1 - v q_y \sigma_2 + m\sigma_3, \tag{2}$$

where $\mathbf{q} = (q_x, q_y)$ is the momentum relative to the Dirac points, $v = \sqrt{3}t/2$ is the group velocity, and $\tau = \pm 1$ is the valley index. Evidently, the effective Hamiltonian is composed of a conventional Dirac Hamiltonian, as in the graphene case, and a unique valley-dependent non-Hermitian term, $\tau 3\sqrt{3}i\delta\sigma_0$, which is the key to the formation of

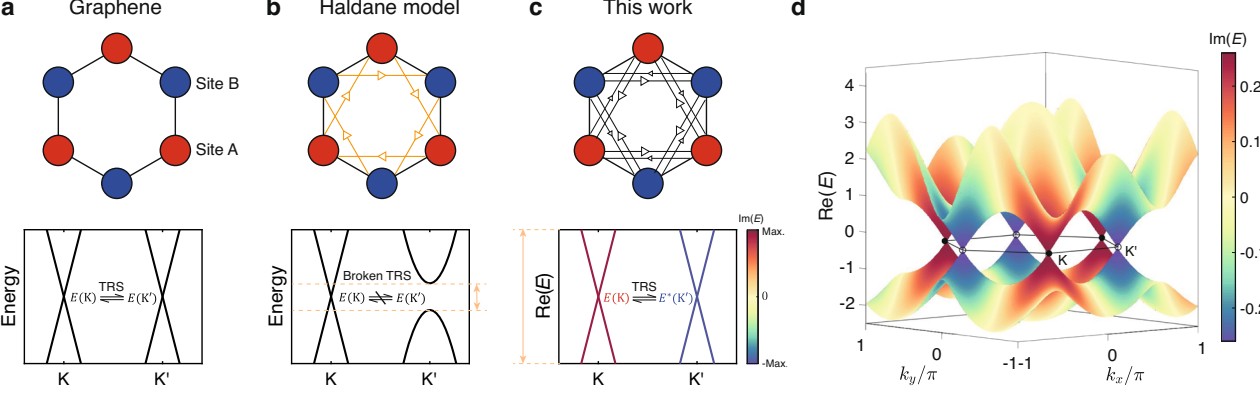

**Fig. 1 | Model for realizing non-Hermitian Dirac quasiparticles. a–c** Upper panels: Tight-binding model of conventional graphene (**a**), the Haldane model (**b**), and this work (**c**). The red (blue) circles represent the sublattices with on-site energy $m$ ($-m$) and the black lines indicate the nearest-neighbor couplings with strength $t$. The Orange arrows in (**b**) indicate the next-nearest-neighbor (NNN) couplings $t_2 e^{i\phi}$ ($t_2 e^{-i\phi}$) along (opposite to) the arrow. The larger (smaller) black arrows in (**c**) denote the NNN couplings $t_2 + \delta$ ($t_2 - \delta$). Lower panels: Schematics of the Dirac cone dispersion of the graphene model (**a**), the Haldane model (**b**), and the non-Hermitian model in this work (**c**). For graphene, the K and K′ valleys are related to each other by the time-reversal

symmetry (TRS). For the Haldane model, the TRS breaking leads to a single Dirac cone in the K valley in the energy range within the bandgap of the K′ valley, as denoted by the orange dashed lines. In this work, a complex conjugate pair of Dirac cones at the K and K′ valleys are related to one another by the TRS, with colors denoting the imaginary parts of the eigenvalues. In the long time limit, only states at the K valley will survive due to the sharp contrast in lifetime between the two valleys, leading to an effective single Dirac cone behavior over a large energy range as indicated by the orange dashed lines. **d** Complex band structure of our proposed model for $m = 0$, $t = 1$, $t_2 = 0.1$, and $\delta = 0.1$, with the colors showing the imaginary parts of the eigenvalues.

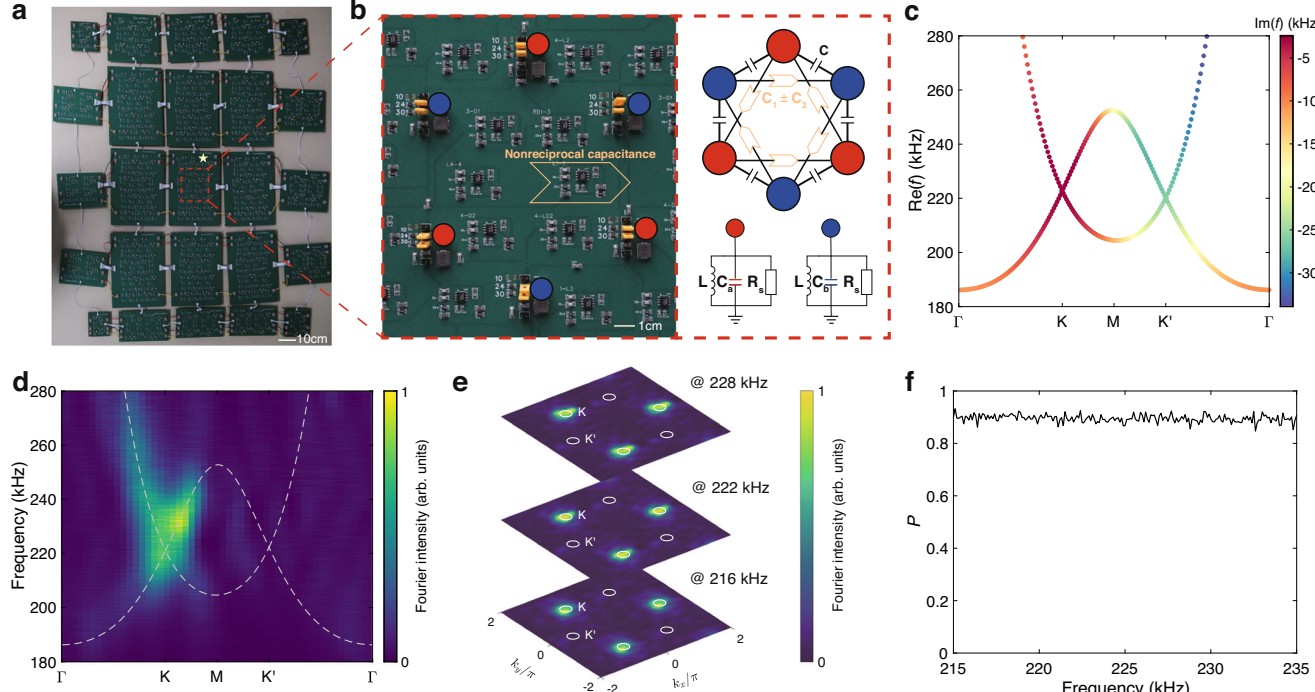

**Fig. 2 | Observation of massless non-Hermitian Dirac quasiparticles. a** Photo of the fabricated circuit sample. The dashed box outlines a hexagonal cell. **b** Left panel: The enlarged view of the hexagonal cell in (**a**). The orange arrow denotes the components for realizing the nonreciprocal couplings. Right panel: The schematic diagram of the designed circuit realizing the hexagonal unit cell. The next-neighbor couplings are achieved through capacitors $C$. The nonreciprocal next-nearest-neighbor couplings are achieved via nonreciprocal capacitance ($C_1 \pm C_2$). Sublattices A/B are grounded with an LC resonator circuit containing a capacitor $C_{a/b}$, an inductor $L$, and a resistor $R_s$. The resistor $R_s$ is added as the global dissipation for circuit stability. **c** Calculated bulk dispersion of the circuit lattice with parameters $C_a = C_b = C = 100$ nF, $C_1 = C_2 = 10$ nF, $L = 1.043 \, \mu$H and $R_s = 13.3 \, \Omega$. The colors denote the imaginary part of the eigenfrequencies. **d**, **e** Experimentally measured dispersions of the circuit lattice along high-symmetry lines (**d**) and in the two-dimensional momentum space at different frequencies (**e**). The colors represent the normalized Fourier intensity. The dashed lines in (**d**) represent the theoretical results. **f** Plot of the measured valley polarization against the driving frequency. The circles in (**e**) denote the integral region for calculating the valley polarization.

the non-Hermitian Dirac cones[22]. Note that the valley-dependent non-Hermitian term is enforced by the TRS, which is still preserved after introducing the non-Hermitian NNN coupling (Fig. 1c). Under the TRS, the effective Hamiltonians at two valleys are related as $TH_K^{tb}(\mathbf{q})T^{-1} = H_{K'}^{tb}(-\mathbf{q})$. Since $T = \mathcal{K}$ is the complex conjugation operation, the eigenvalues at the two valleys must have opposite imaginary parts (see Methods for detailed symmetry analysis).

It is helpful to compare this model with the Haldane model to gain more insights into its physics. In both models, the two Dirac cones have nonidentical dispersions in either the real or imaginary part. In the Haldane model, this is caused by the nonequal Dirac masses induced by the complex NNN coupling, corresponding to a real gauge field, and an on-site mass detuning. While in our case, the imaginary eigenvalue contrasting results from the nonreciprocal NNN coupling that can be interpreted as a consequence of an imaginary gauge field[23]. Thus, our model can also exhibit single Dirac cone physics in the long time limit, using non-Hermiticity instead of TRS breaking. Moreover, the energy range for having states in a single valley is much larger in our case as the imaginary eigenvalue contrasting goes far beyond the low-energy limit (see Fig. 1c). At the boundary, our model can support topological modes protected by the valley Chern number, which also have valley-dependent imaginary eigenvalues and effectively show a unidirectional propagation akin the chiral edge modes in the Haldane model (Figs. 4 and 5).

**Observation of massless non-Hermitian Dirac cones**
We experimentally study this tight-binding model using an electric circuit lattice, which has been a popular platform for studying complex physical models because of the wide variety of electrical elements and

highly flexible connectivity it offers[24-30]. Due to the size limit of the printed circuit board (PCB) fabrication, we segment the entire sample into pieces and interconnect them using connectors (see Methods for circuit details), as shown in Fig. 2a. The zoomed-in image in the left panel of Fig. 2b illustrates the designed circuit realizing one hexagonal cell with six lattice sites, and its schematic diagram is shown in the right panel. The nonreciprocal NNN couplings are achieved through nonreciprocal capacitances ($C_1 \pm C_2$, denoted with orange arrows), which are realized by connecting a negative impedance converter with current inversion (INIC) in parallel with a capacitor $C_1$ (see Supplementary Information for more details)[27]. The nearest-neighbor couplings are realized via capacitors $C$ and the two sublattices are grounded with an LC resonator circuit containing a capacitor $C_{a/b}$ and an inductor $L$. Resistors $R_s$ are added from each node to the ground to avoid instabilities (see Methods for detailed analysis of circuit stability). In our designed circuit, values of $C_a$, $C_b$, and $R_s$ can be modified with switches composed of two-pin headers. Besides, additional components are added at the boundaries of the circuits for all the samples in this work, including suitable capacitors for obtaining the same diagonal elements of the open circuit Hamiltonian and absorbing resistors $R_a = 30 \, \Omega$ to eliminate the reflections (see Supplementary Information for more details). With the absorbing resistors, the non-Hermitian skin effect induced by the boundaries is suppressed and we will work on the conventional Brillouin zone throughout this work.

The Kirchhoff's equations of the circuit are given by (see Supplementary Information for detailed derivations)

$$H^c(\mathbf{k})\mathbf{V} = \left(\frac{1}{\omega^2 L} - i\frac{1}{\omega R_s}\right)\mathbf{V}, \quad (3)$$

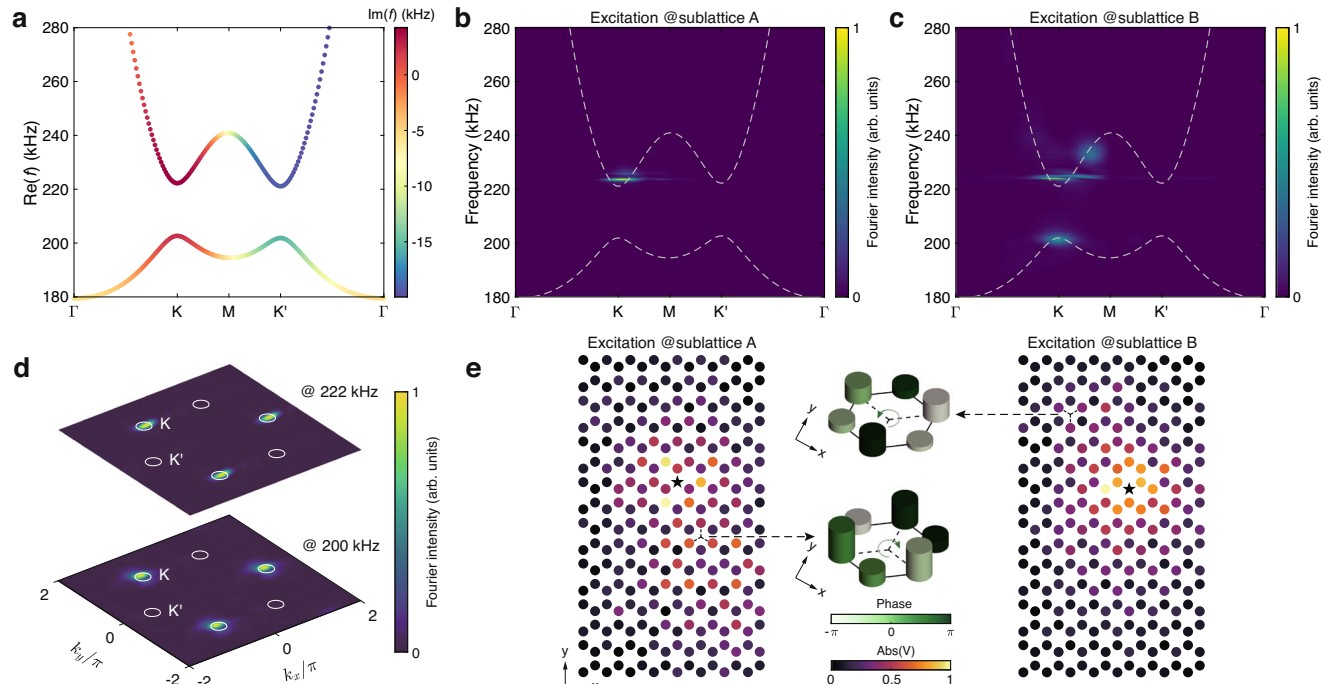

**Fig. 3 | Observation of massive non-Hermitian Dirac quasiparticles. a** Calculated bulk dispersion of the circuit lattice with parameters $C_a = 100$ nF, $C_b = 200$ nF, $C = 100$ nF, $C_1 = C_2 = 10$ nF, $L = 1.043\,\mu$H and $R_s = 30\,\Omega$. The colors denote the imaginary part of the eigenfrequencies. **b, c** Experimentally measured dispersions of the circuit lattice along high-symmetry lines when exciting at the sublattice A (**b**) and the sublattice B (**c**), respectively. The colors represent the normalized Fourier intensity, and the dashed lines represent the theoretical results. **d** Experimentally measured dispersion of the circuit lattice in the two-dimensional momentum

space. The measurements are taken at 222 kHz (upper panel) and 200 kHz (lower panel) with excitations at the sublattices A and B, respectively. **e** Measured voltage amplitudes at 222 kHz (left panel) and 200 kHz (right panel) with excitations at the sublattices A and B, respectively. The black star denotes the source. Zoomed-in bar diagrams plot the amplitudes (heights) and phases (colors) distribution in a hexagonal cell and demonstrate the clockwise and anti-clockwise phase vortices at sublattices A (lower panel) and B (upper panel).

where $\mathbf{V} = [v_a, v_b]^T$ are the node voltages on the two sublattices and $\omega$ is the angular frequency. $H^c(\mathbf{k}) = \sum_{i=0,1,2,3} h_i^c(\mathbf{k})\sigma_i$ is the Hamiltonian of the circuit, with $h_0^c = \sum_{i=1,2,3} -2[C_1\cos(\mathbf{k}\cdot\mathbf{a}_i) - iC_2\sin(\mathbf{k}\cdot\mathbf{a}_i)] + 3C + 6C_1 + (C_a + C_b)/2$, $h_1^c = -C[1 + \cos(\mathbf{k}\cdot\mathbf{a}_2) + \cos(\mathbf{k}\cdot\mathbf{a}_3)]$, $h_2^c = -C[\sin(\mathbf{k}\cdot\mathbf{a}_2) - \sin(\mathbf{k}\cdot\mathbf{a}_3)]$, and $h_3^c = (C_a - C_b)/2$. $H^c$ has a similar form to the tight-binding Hamiltonian (see Eq. (1)), except that $H^c$ contains a global offset ($3C + 6C_1$), which only globally shifts the eigenvalues along the real axis but has no influences on the eigenstates. The circuit dispersion (i.e., the relationship between frequency and momentum) can be obtained by first solving Eq. (3) and then mapping the eigenvalue $E(\omega)$ to the frequency regime through $E(\omega) = 1/\omega^2 L - i/\omega R_s$.

We first implement a sample with massless non-Hermitian Dirac cones (i.e., $m = 0$ in the tight-binding model). Figure 2c shows the calculated dispersion with parameters $C_a = C_b = C = 100$ nF, $C_1 = C_2 = 10$ nF, $L = 1.043\,\mu$H, and $R_s = 13.3\,\Omega$. Consistent with the results in the tight-binding model, two Dirac cones with different imaginary eigenfrequencies can be clearly seen at the K and K' valleys. To verify the designed circuit, we measure the steady-state voltage distributions under a single-site excitation in the bulk using a vector network analyzer and reconstruct the bulk dispersion by Fourier transforms (see Methods for experimental details). As shown in Fig. 2d, e, the Fourier intensity is highly concentrated around the K valley, with negligible distribution at the other valley, suggesting that only a single Dirac cone survives in the steady state. Notably, this phenomenon is guaranteed by the non-Hermitian dispersion and persists even when the frequency is away from the Dirac points. The valley polarization is used here to quantify the single Dirac cone performance, which is defined as[31]

$$P = \frac{F_K - F_{K'}}{F_K + F_{K'}}, \qquad (4)$$

where $F_K$ and $F_{K'}$ denote the integral of the Fourier intensity around the K and K' valleys, respectively. The integral regions around different valleys are marked with circles in Fig. 2e. The polarization $P \in [-1, 1]$, with $P = 1$ if the modes lie fully in the K valley and $P = -1$ if they lie fully in the K' valley. As shown in Fig. 2f, large valley polarizations $P \sim 1$ exist over a wide frequency range, demonstrating the large bandwidth of having Dirac quasiparticles in a single valley.

## Observation of massive non-Hermitian Dirac cones

We then introduce an on-site mass detuning (i.e., $C_a = 100$ nF and $C_b = 200$ nF) to study the physics in the massive case. This results in a bandgap (Fig. 3a), and opposite orbital magnetic moment and Berry curvature at the two valleys[8,9] (see Supplementary Information for more details), in addition to the original imaginary eigenfrequency contrasting. We excite the circuit at sublattices A and B, respectively, to measure the upper and lower bands, as given in Fig. 3b–d. It is observed that wherever the circuit is excited at sublattice A or B, only the dispersion at the K valley can be seen, similar to the massless case. Due to the significant contrast between imaginary parts of eigenfrequencies, modes with much larger imaginary parts (i.e., those near the band edges) dominate the measured dispersion. Note that the upper band is also excited when we excite at sublattice B since the upper band has a larger imaginary part than the lower band (see Fig. 3a).

In the massive case, the single Dirac cone feature can also be directly judged from the real-space fields. The valley-contrasting orbital magnetic moment dictates that the states at the two valleys exhibit vortices with opposite chirality[8]. When states in only one valley are presented, the steady-state field will carry a phase winding pattern. We indeed find such a feature from our experimental data, as shown in Fig. 3e. The fields at 222 kHz and 200 kHz exhibit opposite phase

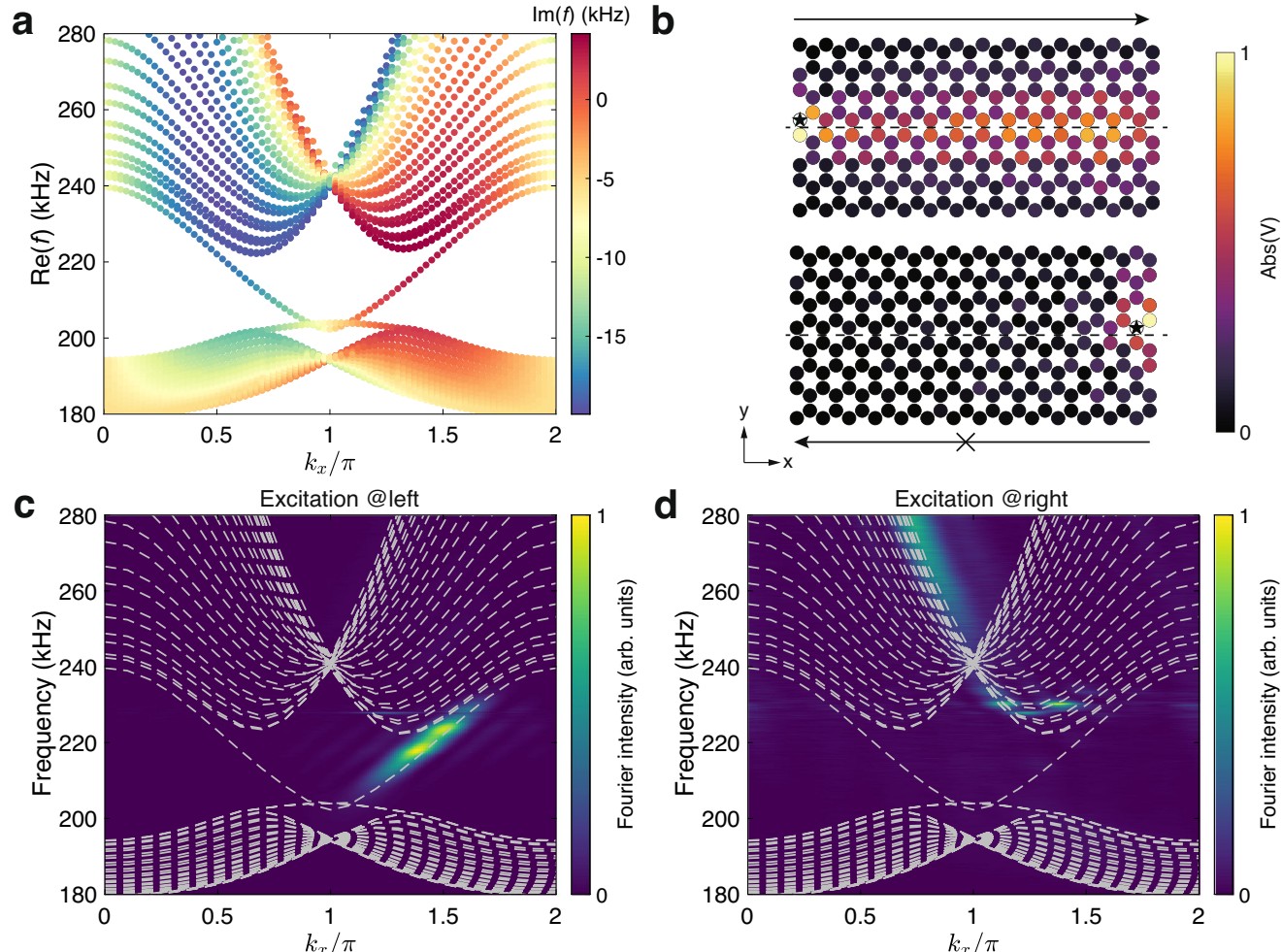

**Fig. 4 | Topological valley kink states with valley-dependent imaginary eigenfrequencies. a** Calculated dispersion of a heterostructure with a zigzag interface constructed by a lattice with $C_a = 200$ nF and $C_b = 100$ nF at the upper domain and a lattice with $C_a = 100$ nF and $C_b = 200$ nF at the lower domain. Other parameters are $C = 100$ nF, $C_1 = C_2 = 10$ nF, $L = 1.043\,\mu$H and $R_s = 30\,\Omega$. The colors denote the imaginary part of eigenfrequencies. **b** Measured voltage field distributions at 215 kHz with excitations at the left (upper panel) and right (lower panel) sides. The black star denotes the source and the dashed lines mark the location of the zigzag interface. **c**, **d** Experimentally measured edge dispersion from Fourier transform of the fields at the interface with excitations at the left (**c**) and right (**d**) sides, respectively. The colors represent the normalized Fourier intensity and the dashed lines represent the theoretical results.

windings, consistent with the fact that the upper and lower bands have opposite orbital magnetic moment. We note that, thanks to the auto valley selection enabled by non-Hermiticity, the vortex states are excited by a simple single-site excitation. While in previous studies, the generation of such vortex states in Hermitian systems requires complex sources like a phased array[32].

**Observation of non-Hermitian topological valley kink states**

Next, we turn to look at the boundary physics originating from the non-Hermitian Dirac cones in the bulk. It is known that an interface between two domains with opposite Dirac masses can support gapless valley kink states protected by the valley Chern numbers[33–35]. As Eq. (2) only differs from a conventional Dirac Hamiltonian by a uniform imaginary part, our model should preserve the non-trivial valley Chern numbers (see Supplementary Information for more details) and enjoy the same boundary topology. Our numerical calculation on a circuit lattice with a zigzag interface indeed finds such valley kink states, but with additional valley-dependent imaginary eigenfrequencies (Fig. 4a), which also agrees well with the tight-binding results as shown in Supplementary Information.

When a source is attached to the left side, the kink state is excited and propagates along the interface (Fig. 4b, upper panel). However, when the source is relocated to the right end, the field is localized around

the excitation without any propagation signature (Fig. 4b, lower panel). The corresponding Fourier spectra for these two cases are given in Fig. 4c and d, showing that the right-moving states are successfully excited while the left-moving ones are not due to their significant decay. This unidirectional propagation behavior is uniquely caused by the non-Hermiticity and is essentially different from the conventional propagation of chiral edge states induced by a nonzero Chern number. Besides, our non-Hermitian valley kink states are also distinct from those with on-site gain/loss that do not feature valley-dependent lifetime and thus cannot exhibit unidirectional propagation[36]. From the non-Hermitian topology point of view, this interface mimics the classic Hatano-Nelson model with a non-trivial point-gap topology[23] (see Supplementary Information for more discussions).

Finally, we demonstrate that the non-Hermitian valley kink states are more robust than their Hermitian counterparts. This is basically because the difference in the imaginary eigenfrequencies of the kink states in the two valleys makes it harder for inter-valley scattering to occur. To see this, we consider the armchair interface on which the projections of the two valleys overlap. In the Hermitian case, the valley kink states are usually gapped due to the inter-valley coupling[37]. However, the imaginary eigenfrequencies contrast in our model can make the kink states decouple and thus restore gapless dispersion.

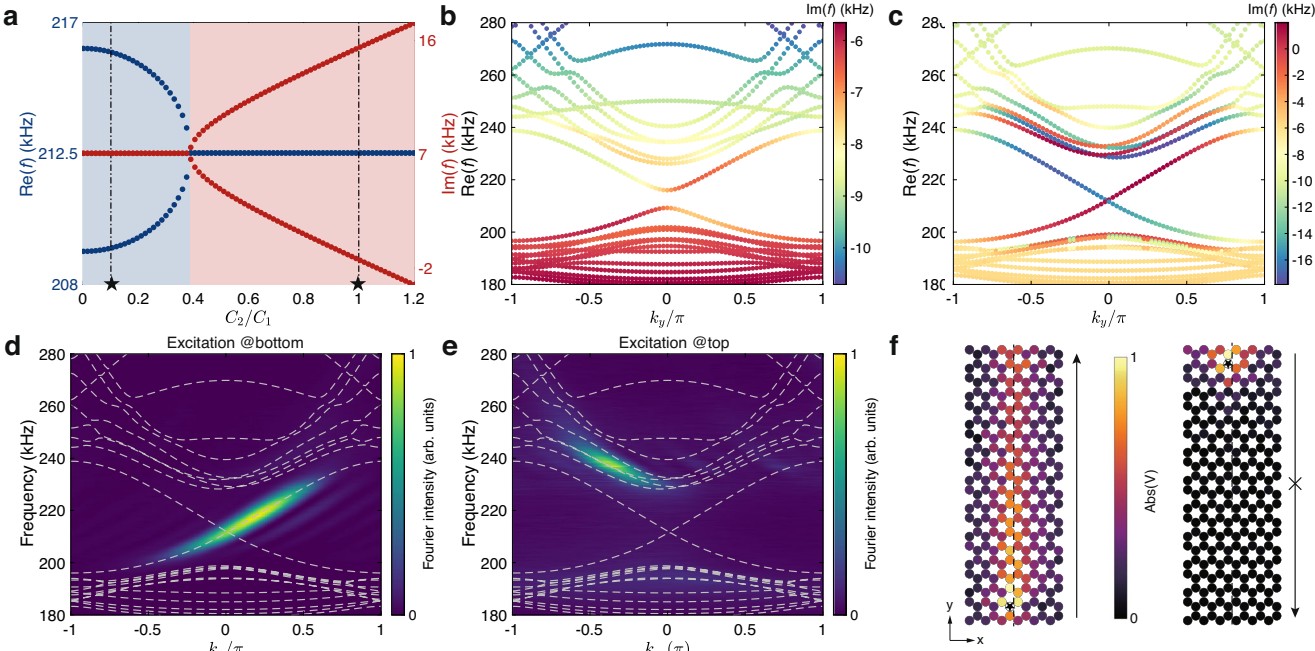

**Fig. 5 | Suppressed inter-valley scattering at the armchair interface. a** Plots of the real (blue dots) and imaginary (red dots) parts of the valley kink states' eigenfrequencies at $k_y = 0$ of a heterostructure with an armchair interface against the ratio between the nonreciprocal part $C_2$ and the reciprocal part $C_1$ of the NNN coupling. The blue and red regions denote the parity-time symmetric and parity-time symmetry-broken phases, respectively. **b, c** Calculated dispersions for the two cases denoted by the black stars in (**a**) ($C_2/C_1 = 0.1$ for (**b**) and $C_2/C_1 = 1$ for (**c**)). The colors denote the imaginary part of the eigenfrequencies. Circuit parameters are

$C = 100$ nF, $C_1 = 10$ nF, $L = 1.043\,\mu$H, and $R_s = 30\,\Omega$. **d, e** Experimentally measured edge dispersions from Fourier transform of the fields at the interface with excitations at the bottom (**d**) and top (**e**) sides, respectively. The colors represent the normalized Fourier intensity and the dashed lines represent the theoretical results. **f** Measured voltage field distributions at 215 kHz with excitations at the bottom (left panel) and top (right panel). The black star denotes the source and the dashed lines mark the location of the armchair interface.

Figure 5a plots the evolution of the eigenfrequencies of the kink states at $k_y = 0$ (where the kink states cross each other) as a function of the dimensionless non-Hermitian parameter $C_2/C_1$. As can be seen, a gap is present in the Hermitian case (i.e., $C_2 = 0$). As $C_2/C_1$ increases, the gap size decreases and eventually becomes zero after a parity-time phase transition. Figure 5b, c show the calculated dispersions before and after the phase transition, respectively. It can be clearly seen that the edge states after the phase transition are gapless and have contrasting imaginary eigenfrequencies, similar to the zigzag case. These results are consistent with the calculations from the tight-binding model (see Supplementary Information for more details). The measured (colormaps) and calculated (dashed lines) projected dispersions for the armchair valley kink states are shown in Fig. 5d, e, corresponding to the excitations at the bottom and top, respectively. Again, a unidirectional propagation is observed, where the kink states can only propagate upwards but not downwards (Fig. 5f).

## Discussion

In summary, we have observed non-Hermitian Dirac cones in a graphene-like model with nonreciprocal NNN couplings. Protected by the TRS, Dirac quasiparticles in different valleys acquire highly contrasting lifetimes, leading to intriguing non-Hermitian valley contrasting physics. Using circuiting experiments, we demonstrate the effective single Dirac cone behavior of our model, which allows for auto valley selection and vortex state generation. At the boundary, topological valley kink states are found to inherit the non-Hermitian physics in the bulk and exhibit unidirectional propagation and suppressed inter-valley scattering that are not seen in the Hermitian case, which could be used for robust logic gates and compact circuit integration. These results demonstrate that Dirac quasiparticles can not only exist in realistic non-Hermitian settings but also have profound physical implications.

Our work takes the first step in studying non-Hermitian quasiparticles and extends the current studies on non-Hermitian topological semimetal phases which are mostly focused on exceptional points and conventional Hermitian band degeneracies. There are many efforts to be made along this direction. At the fundamental level, it would be interesting to search for other types of non-Hermitian quasiparticles and their interactions with external fields[38,39], especially non-Hermitian Weyl fermions, which have been predicted to host exotic three-dimensional non-Hermitian physics[22,40–43]. Practically, it is highly desired to extend our results to wave (e.g., microwave, optical and acoustic) and even electronic systems, where various valley-based applications may be further developed.

## Methods
### Symmetry analysis

The non-Hermiticity of our graphene model is introduced by the real but nonreciprocal NNN couplings $t_2 \pm \delta$ shown in Fig. 1c. These couplings preserve the TRS $T = \mathcal{K}$, but break the inversion symmetry. Hence, our model respects TRS as

$$TH^{\text{tb}}(\mathbf{k})T^{-1} = H^{\text{tb}}(-\mathbf{k}),$$

where $T = \mathcal{K}$ is the complex conjugation operation, but it does not preserve the inversion symmetry $I = \sigma_1$ as

$$IH^{\text{tb}}(\mathbf{k})I^{-1} \neq H^{\text{tb}}(-\mathbf{k}).$$

Next, let us focus on the effective Hamiltonians at the two valleys, which are linear approximations of the original tight-binding model,

$$H_K^{\text{tb}}(\mathbf{q}) \approx H^{\text{tb}}(K+\mathbf{q}), \quad H_{K'}^{\text{tb}}(\mathbf{q}) \approx H^{\text{tb}}(K'+\mathbf{q}).$$

Since the original Hamiltonian respects the time-reversal symmetry, i.e., $TH^{tb}(K+\mathbf{q})T^{-1}=H^{tb}(-K-\mathbf{q})=H^{tb}(K'-\mathbf{q})$, the two effective Hamiltonians are related by

$$TH_{K}^{tb}(\mathbf{q})T^{-1}=H_{K'}^{tb}(-\mathbf{q}).$$

Hence, the energy eigenvalues at the two valleys must have opposite imaginary parts.

## Sample design and fabrication

We utilize EasyEDA for the design and optimization of our electric circuits, where the PCB composition, stack-up layout, internal layer and grounding design are suitably engineered. Each PCB is implemented on FR4 and consists of six layers to arrange the complex conductor. The ground layers are placed in the gap between the source and signal layers to avoid their coupling. To reduce parasitic inductance, all signal traces are designed with a relatively large width of 0.508 mm. Source traces are even wider, at 0.635 mm, to ensure the safety and stability of the power supply. Additionally, the spacing between electronic components is set to be larger than 1.5 mm to prevent spurious inductive coupling.

Due to the PCB fabrication size limitations, we divide the overall design into smaller pieces (see Fig. 2a). With careful arrangement, these pieces can realize various configurations, including massless bulk, massive bulk, zigzag interface, and armchair interface.

## Circuit details

To minimize the influence of parametric disorder, all circuit elements are pre-characterized at 220 kHz to have relatively small tolerances, as given by the following. The parameter values of the inductor $L$ is 1.043 $\mu$H with $\pm$ 3% tolerance and 28 m$\Omega$ series resistance. Capacitors used to realize $C_a$, $C_b$, C, $C_1$, and $C_2$ include 100 nF and 10 nF, each with $\pm$ 1% tolerance. Resistors used to realize $R_s$ and $R_a$ include 24 $\Omega$ and 30 $\Omega$ resistors, each with $\pm$ 1% tolerance. The operational amplifier is LT1363. The impedance $Z_0$ in the INIC module is realized by connecting a resistor $R_0$ = 20 $\Omega$ with 1% tolerance and a capacitor $C_0$ = 100 nF with $\pm$5% tolerance in parallel. In measurement, a pair of filter capacitors (2.2 $\mu$F with $\pm$ 10% tolerance and 2 pF with $\pm$ 5% tolerance) are connected in parallel with the output of DC supply (KORAD KA3005DS) to provide $\pm$ 5 V DC voltages for the optional amplifiers, ensuring minimized ripple current.

## Circuit stability

In the implementation of the INIC, we employed the unit-gain stable operational amplifier model LT1363. The realistic operational amplifiers have some non-ideal characteristics, including finite gain, finite bandwidth, and phase delay. These non-ideal characteristics will affect the frequency response of the INIC, thus affecting the phase margin. To sufficiently increase the gain and phase margins of the INIC, we implement the impedance $Z_0$ (see Supplementary Information) by connecting a resistor $R_0$ = 20 $\Omega$ in parallel with a capacitor $C_0$ = 100 nF.

As the INIC is an active, nonreciprocal circuit element, it pumps energy into the system. This can lead to eigenstates with positive imaginary components in their eigenfrequencies, which manifest as exponential growth in magnitude over time. Consequently, energy builds up in the system until the operational amplifier exhibits nonlinear saturation effects and ceases to function properly. To prevent the accumulation of self-sustained energy gains, resistors $R_s$ are added from each node to the ground. These resistors are designed to dissipate the desired amount of power and dampen all modes uniformly. The value for $R_s$ should be carefully selected: it must be sufficiently small to prevent instabilities but large enough to avoid oversized damping, which localizes the voltage response and likewise decreases the measurement accuracy. Numerical computations of the eigenfrequencies indicate that $R_s \leq 16 \Omega$ is required to eliminate all eigenstate

divergences. In practical experimental implementations, parasitic resistances and the different eigenstate distributions across different samples contribute to stabilizing the eigenstates and further increase the necessary $R_s$. For the massless bulk, we choose $R_s$ = 13.3 $\Omega$, while for the massive bulk, as well as the zigzag and armchair interfaces, we use $R_s$ = 30 $\Omega$.

## Experimental measurements

We first use jumper wires to connect circuit pieces to realize a specific structure such as the massless bulk as shown in Fig. 2a. The experiment measurement setup is shown in Fig. S7 in Supplementary Information. The voltage is measured by a two-port network vector analyzer (KEYSIGHT N9927A). We place port 1 at the excitation node and use port 2 to measure the voltage responses $U_{x,y}$ of all the other nodes. By applying the Fourier transformation to $U_{x,y}$, we obtain the dispersion diagrams. The DC supplies (KORAD KA3005DS) are used to provide $\pm$5 V DC voltages for the optional amplifiers. In the measurements, the output power is $-$3 dBm, the frequency sweep size is 50 Hz, and the intermediate frequency bandwidth (IFBW) is 3 kHz, which all help to reduce measurement errors.

## Data availability

All the data supporting this study are available in the paper and Supplementary Information. Additional data related to this paper are available from the corresponding authors upon request.

## Code availability

The codes that support the findings of this study are available from the corresponding authors upon request.

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

## Acknowledgements

We are grateful to Yidong Chong and Baile Zhang for fruitful discussions. X.X., F.M., Z.D., Y.D., H.C. and F.G. are supported by the Key Research and Development Program of the Ministry of Science and Technology under grants No. 2022YFA1404902, 2022YFA1404704 and 2022YFA1405200, the National Natural Science Foundation of China (NNSFC) under grants No. 62171406, 11961141010 and 61975176, the Zhejiang Provincial Natural Science Foundation under grant No. R25F010016, the Key Research and Development Program of Zhejiang Province under grant No. 2022C01036, and the Fundamental Research Funds for the Central Universities. H.X. acknowledges the support from the National Natural Science Foundation of China (grant No. 62401491) and the Chinese University of Hong Kong (grants No. 4053675, 4937205 and 4937206). W.B.R. is supported by the RGC Postdoctoral Fellowship (Ref. No. PDFS2223-7S05). Y.X.Z. is supported by the GRF of Hong Kong (Grant No. 17301224).

## Author contributions

F.G. and H.X. conceived the idea. X.X., W.B.R., Y.X.Z., and H.X. did the theoretical analysis. X.X., F.M., and W.X. designed the sample. X.X., F.M., Z.D., and Y.D. conducted the experiments. X.X., W.B.R., Y.X.Z., H.C., F.G., and H.X. wrote the manuscript with input from all authors. F.G. and H.X. supervised the project.

## Competing interests

The authors declare no competing interests.
