## [Transparent Peer Review file · Nature Communications]

Non-Hermitian Dirac cones with valley-dependent lifetimes

Corresponding Author: Dr Haoran Xue

Version 0:

Reviewer comments:

Reviewer #1

(Remarks to the Author)

This paper reports on an experimental realization of a non-Hermitian extension of the graphene lattice with a complex conjugate pair of Dirac cones, using an electrical circuit lattice. As demonstrated in the earlier theory paper [22], the Dirac quasiparticles in this model are governed by non-Hermitian Dirac Hamiltonians with a valley-dependent background imaginary term, which is confirmed by measuring the steady-state voltage distributions under a single-site excitation. Moreover, the valley kink states with valley-locked lifetime are also demonstrated.

The paper is theoretically sound and the experimental results are clearly presented. It is an interesting work which provides experimental demonstration of a new type of non-Hermitian Dirac cones and valley filter. There are a few points that I believe the authors need to address.

- (1) I think that the title sounds rather general. Despite the essential differences, the “non-Hermitian Dirac cone” have been termed in various work.
- (2) In the lower panel of Fig. 1c, it is confused that “the sharp imaginary eigenvalue contrast leads to an effective single Dirac cone with a large bandwidth.”
- (3) It seems that the eigenspectrum in Fig. 1d is not symmetric about $\text{Re}(E)=0$. Is the chiral/sublattice symmetry broken by introducing the nonreciprocal NNN couplings? In addition, because of the role of the frequency in circuit system (as a tuning parameter), the connection between the resonant frequency spectrum in Fig. 2c and the eigenspectrum in Fig. 1d should be clarified.
- (4) The non-Hermitian valley edge states have been also studied in reference [Phys. Rev. Lett. 120, 246601 (2018)]. Is there any difference between the transport behavior of the valley kink states here and valley edge states in this paper?
- (5) The references Phys. Rev. Lett. 130, 103602 (2023) and Nat. Commun. 15, 1798 (2024), which introduce a “non-Hermitian Dirac Hamiltonian” under a pseudomagnetic field and realize it using non-Hermitian electric circuits, should be cited.

Reviewer #2

(Remarks to the Author)

In this paper, the authors have proposed the two-dimensional non-Hermitian Dirac cones. From the theoretical side, as a concrete example, the authors have proposed the honeycomb-lattice tight-binding model with non-reciprocal next-nearest-neighbor hopping. In this model, the non-reciprocity gives rise to the momentum-dependent imaginary part of the energy, thus the amplifying (decaying) Dirac cone appears in K (K') point. They have further realized this model in the electric circuit, where the non-reciprocal hopping is implemented by using the negative impedance converter with current inversion (INIC). They have observed the valley-selective intensity for the bulk, topological kink states with an effectively unidirectional nature.

The theoretical analysis presented in this paper is solid and the experimental results seem sound. Although the model looks quite simple, the Dirac cones with valley-dependent life time have not been proposed yet, to my knowledge. However, I have concerns about the broad impact of this result when considering the publication in Nature Communications, which I list

below.

(i) The authors mention that extending this finding to the various waves and electronic systems is desirable. Still, I think that the non-reciprocal hopping is not widely feasible in physical systems. To my knowledge, the electric circuits with INIC are the only example that can realize the non-reciprocal hopping experimentally. So I wonder whether the non-Hermitian Dirac cones proposed in this paper have broad feasibility in various physical systems other than the electric circuit.

(ii) I find that the non-Hermitian Dirac cones and the topological kink states are of fundamental interest. However, it is not clear to me what are the possible applications when implementing them in the electric circuit. I think that the authors need to present the motivation to realize the non-Hermitian Dirac cones in the electric circuit.

In addition to the above two points, I also would like the authors to consider the following questions and comments.

(iii) In the caption of Fig. 1, the authors have stated “The bandwidth for the single Dirac cone is determined by the gap size of the K' valley, ...” regarding Fig. 1b, but the meaning of “the bandwidth of the single Dirac cone” is not clearly stated. Naively, the bandwidth of the Dirac cone is determined by to what extent the linear approximation of the Hamiltonian works near the Dirac point and has nothing to do with the information of the opposite valley. I suppose that the authors mean as follows: Within the energy range in the band gap of K' valley, the only existing eigenmodes are the massless Dirac cone at K point. So I would like the authors to clarify the meaning of the bandwidth of the single Dirac cone in this context.

(iv) In the electric circuits, it is possible to realize the periodic boundary condition by connecting the nodes at both ends (in each direction) directly, while the actual circuit in Fig. 2a seems to correspond to the open boundary condition. I think assigning the periodic boundary condition is advantageous in that the obtained spectrum can be directly compared to the band structure in the momentum-space picture. I wonder why the authors implement the open boundary condition rather than the periodic boundary condition.

(v) In Figs. 2c, 3a, 4a, 5b and 5c, the authors plotted the “calculated dispersion”, where the vertical axis is $\text{Re}(f)$, but it is not clear to me how this quantity is related to the eigenvalue equation of Eq. (3). It seems to me that the plotted curves correspond to the real part of the eigenvalues of $H^c(k)$ [i.e., $E(\omega)$], but the eigenvalue is not the frequency (i.e., ω) itself, as the authors state in the same paragraph as Eq. (3). Thus, I would like to authors to clarify what is the definition of $\text{Re}(f)$ and how it is related to $E(\omega)$ or ω in Eq. (3).

(vi) In the right column of page 6, the authors have stated “... thus restore gapless transport”. I think the word “gapless transport” is ambiguous, so the authors will need to clarify what it means. In the electric circuit, the kink states give rise to characteristic voltage distribution but their relation to the transport phenomena is unclear.

Reviewer #3

(Remarks to the Author)

The paper by Xinrong Xie et al. is an experimental work on the realization of non-Hermitian Dirac cones in circuits. By introducing non-reciprocal next-nearest-neighbor couplings via circuit elements, the authors first experimentally demonstrate the massless and massive non-Hermitian Dirac cones at K and K' points. Afterward, topological valley kink states of a zigzag interface and parity-time transition of kink states at an armchair interface have been carefully investigated. The results are clear, and therefore, I would suggest its publication in Nature Communications if the authors can address the following issues:

(1) It is known that when a non-Hermitian system has skin modes, all the properties, including topological invariants, band dispersions, and so on, shall be investigated on the generalized Brillouin zone (BZ) but not on the BZ. From my understanding (for example, see Fig. S6), the system under consideration does not have skin modes and thus can focus on the BZ, which is the current approach used throughout the manuscript. However, can the authors clearly prove and demonstrate this issue in the main text?

(2) All the experimental results, especially the bulk ones, can hardly tell the dispersions solely. Does such an issue come from sample sizes, intrinsic losses, or some other systematic reasons? Please discuss this issue and possible improvements in the main text.

(3) Figure S6(a): Can the authors clearly identify which regions inside the real energy line gap have non-zero winding numbers (or, equivalently, point gaps nontrivially open), especially the region near $\text{Re}(E)=0.2$?

(4) If the authors perform similar measurements in Figs. 5(d-f) on Fig. 5(b), how about the experimental results?

Version 1:

Reviewer comments:

Reviewer #1

(Remarks to the Author)

I thank the authors for their comprehensive response to the comments on their manuscript. I find their revisions sufficient to address my remarks, so I recommend it to be published in Nature Communications.

Reviewer #2

(Remarks to the Author)

The authors have considered the questions and comments I raised in the previous report and they have revised the manuscript and the supplementary information. They have provided a convincing explanation that their finding of the non-Hermitian Dirac cones with the valley-dependent life time will have broad potential applications not only for the electric circuits but also for various other physical setups. Therefore, I recommend the publication of this manuscript in Nature Communications.

Reviewer #3

(Remarks to the Author)

The authors have addressed my concerns, so I suggest its publication in its current form now.

Response Letter to Reviewers

We are grateful for the constructive comments on this manuscript (NCOMMS-24-62468-T) from all reviewers. In the text below, the comments from each reviewer are quoted in *blue italics*, and followed by our response. We have also revised the manuscript and the Supplementary Information accordingly, and these updates are highlighted in **red** in those files. In the text below, references to these updates are also highlighted in **red**.

Reviewer #1

General comments from Reviewer #1:

This paper reports on an experimental realization of a non-Hermitian extension of the graphene lattice with a complex conjugate pair of Dirac cones, using an electrical circuit lattice. As demonstrated in the earlier theory paper [22], the Dirac quasiparticles in this model are governed by non-Hermitian Dirac Hamiltonians with a valley-dependent background imaginary term, which is confirmed by measuring the steady-state voltage distributions under a single-site excitation. Moreover, the valley kink states with valley-locked lifetime are also demonstrated.

The paper is theoretically sound and the experimental results are clearly presented. It is an interesting work which provides experimental demonstration of a new type of non-Hermitian Dirac cones and valley filter. There are a few points that I believe the authors need to address.

Response from Authors:

We thank the reviewer for carefully reading our manuscript and recognizing *the experimental demonstration of a new type of non-Hermitian Dirac cones and valley filter*. We also thank the reviewer for considering that our work is *interesting with sound theoretical analysis and clearly-presented experimental results*. Below, we provide point-by-point responses to the comments raised by the reviewer.

Specific comments from Reviewer #1:

Reviewer #1 Comment 1:

I think that the title sounds rather general. Despite the essential differences, the “non-Hermitian Dirac cone” have been termed in various work.

Response from Authors:

We thank Reviewer #1 for raising this issue. Indeed, “non-Hermitian Dirac cones” have been used to describe Hermitian Dirac cones in non-Hermitian systems in previous works. To avoid

potential confusion and distinguish our work from previous ones, we have changed the title to “Non-Hermitian Dirac cones **with valley-dependent lifetimes**”.

Reviewer #1 Comment 2:

In the lower panel of Fig. 1c, it is confused that “the sharp imaginary eigenvalue contrast leads to an effective single Dirac cone with a large bandwidth.”

Response from Authors:

We thank the referee for raising this question. To better explain this point, we have replaced the term “bandwidth” with a more explicit description and expanded this sentence as “**In the long time limit, only states at the K valley will survive due to the sharp contrast in lifetime between the two valleys, leading to an effective single Dirac cone behavior over a large energy range as indicated by the orange dashed lines.**”

Other related sentences in the main text have also been modified accordingly:

Page 2, caption of Fig. 1: For the Haldane model, the TRS breaking leads to a single Dirac cone in the K valley **in the energy range within the bandgap of the K' valley, as denoted by the orange dashed lines.**

Page 3, left column: **Moreover, the energy range for having states in a single valley is much larger in our case as the imaginary eigenvalue contrasting goes far beyond the low-energy limit (see Fig. 1c).**

Besides, **the text “single Dirac cone” has been moved from the lower panel of Fig. 1c.**

Reviewer #1 Comment 3.1:

It seems that the eigenspectrum in Fig. 1d is not symmetric about $Re(E)=0$. Is the chiral/sublattice symmetry broken by introducing the nonreciprocal NNN couplings?

Response from Authors:

Yes, the chiral/sublattice symmetry is broken by introducing the NNN couplings that connect sites of the same sublattice type. Nevertheless, the essential symmetry for the physics, which is the time-reversal symmetry, is preserved under the NNN couplings.

Reviewer #1 Comment 3.2:

In addition, because of the role of the frequency in circuit system (as a tuning parameter), the connection between the resonant frequency spectrum in Fig. 2c and the eigenspectrum in Fig. 1d should be clarified.

Response from Authors:

As indicated by Eq. (3) in the manuscript, there is a one-to-one correspondence between the eigenvalues in the tight-binding model and the eigenfrequencies in the circuit model: $E(\omega) = 1/\omega^2 L - i/\omega R_s$. The eigenvalues $E(\omega)$ are initially obtained by numerically diagonalizing the Hamiltonian, after which the above equation is solved to retrieve the corresponding eigenfrequencies $f = \omega/2\pi$.

Following the reviewer’s suggestion, we have added one sentence on page 4, right column to clarify this issue, which reads “**The circuit dispersion (i.e., the relationship between frequency and momentum) can be obtained by first solving Eq. (3) and then mapping the eigenvalue $E(\omega)$ to the frequency regime through $E(\omega) = 1/\omega^2 L - i/\omega R_s$ ”.**

Reviewer #1 Comment 4:

The non-Hermitian valley edge states have been also studied in reference [Phys. Rev. Lett. 120, 246601 (2018)]. Is there any difference between the transport behavior of the valley kink states here and valley edge states in this paper?

Response from Authors:

FIG. R1. Comparison of the dispersions between our work and the referred work. Band diagrams of the topological valley interface of (a) our work and (b) in the referred work.

In our work, both types of zigzag domain walls (types A and B) support two topological edge states with valley-dependent imaginary eigenfrequencies, as shown in Fig. R1a. This endows each domain wall with unidirectional transport: the left- and right-moving edge states exhibit

amplification and decay, respectively.

In contrast, the referred work [Phys. Rev. Lett. 120, 246601 (2018)] demonstrates that both types of zigzag domain walls support two topological edge states with the same imaginary eigenfrequencies for both valleys, as shown in Fig. R1b. Consequently, each domain wall in that work exhibits identical transport properties for left- and right-moving states, different from the unidirectional transport observed in our work.

To make the readers aware of the essential difference between the two works, we have added a discussion on page 6, left column in the main text, which reads “**Besides, our non-Hermitian valley kink states are also distinct from those with on-site gain/loss that do not feature valley-dependent lifetime and thus cannot exhibit unidirectional propagation [36]**”.

Reviewer #1 Comment 5:

The references Phys. Rev. Lett. 130, 103602 (2023) and Nat. Commun. 15, 1798 (2024), which introduce a “non-Hermitian Dirac Hamiltonian” under a pseudomagnetic field and realize it using non-Hermitian electric circuits, should be cited.

Response from Authors:

We thank the reviewer for providing the related references. We have cited them in the conclusion part as: “There are many efforts to be made along this direction. At the fundamental level, it would be interesting to search for other types of non-Hermitian quasiparticles **and their interactions with external fields [38, 39]**”.

Reviewer #2

General comments from Reviewer #2:

In this paper, the authors have proposed the two-dimensional non-Hermitian Dirac cones. From the theoretical side, as a concrete example, the authors have proposed the honeycomb-lattice tight-binding model with non-reciprocal next-nearest-neighbor hopping. In this model, the non-reciprocity gives rise to the momentum-dependent imaginary part of the energy, thus the amplifying (decaying) Dirac cone appears in K (K') point. They have further realized this model in the electric circuit, where the non-reciprocal hopping is implemented by using the negative impedance converter with current inversion (INIC). They have observed the valley-selective intensity for the bulk, topological kink states with an effectively unidirectional nature.

The theoretical analysis presented in this paper is solid and the experimental results seem sound. Although the model looks quite simple, the Dirac cones with valley-dependent life time have not been proposed yet, to my knowledge. However, I have concerns about the broad impact of this result when considering the publication in Nature Communications, which I list below.

Response from Authors:

We thank the reviewer for carefully reading our manuscript and recognizing *the Dirac cones with valley-dependent lifetime for the first time*. We also thank the reviewer for considering that *the theoretical analysis is solid and the experimental results seem sound*. The comments from the reviewer help us further clarify and strengthen the significance of our work. We address all the issues point-by-point as follows. Especially, we have provided several concrete designs in other physical systems to make our work of broader interest and impact.

Specific comments from Reviewer #2:

Reviewer #2 Comment 1:

The authors mention that extending this finding to the various waves and electronic systems is desirable. Still, I think that the non-reciprocal hopping is not widely feasible in physical systems. To my knowledge, the electric circuits with INIC are the only example that can realize the non-reciprocal hopping experimentally. So I wonder whether the non-Hermitian Dirac cones proposed in this paper have broad feasibility in various physical systems other than the electric circuit.

Response from Authors:

In addition to electric circuits with INIC, non-reciprocal hopping has been experimentally realized in electronic, photonic, acoustic, and mechanical systems, as listed below:

In electronic platforms, the non-reciprocal coupling has been effectively realized by the non-reciprocity of quantum Hall edge states [Nat. Phys. 20, 395–401 (2024)].

In photonic platforms, the non-reciprocal coupling has been realized in quantum walks [Nat. Phys. 16, 761 (2020)] and in optical fibers by optical gain and loss [Science 368, 311 (2020)].

In acoustic platforms, the non-reciprocal couplings has been realized in ring resonator lattice with loss added to link rings [Nat. Commun. 12, 5377 (2021)] and in coupled acoustic resonators using directional acoustic amplifiers [Nat. Commun. 12, 6297 (2021); Phys. Rev. Lett. 130, 017201 (2023)].

In mechanical platforms, the non-reciprocal couplings has been realized by programmed external actuation of the motors [Proc. Natl. Acad. Sci. U.S.A. 117 (47) 29561-29568 (2020); Nature 608, 50–55 (2022)].

Therefore, we believe that the proposed non-Hermitian Dirac cone model has the potential for implementation in various physical systems beyond electric circuits. In the revised version, we have added two designs (one for sound waves and the other for mechanical vibration) for the realization of our model beyond circuits in Supplementary Information. The corresponding revisions are also given below for the reviewer’s convenience.

S8. Realizing non-Hermitian Dirac cones in other platforms.

In addition to the electric circuits with INIC, there are other ways to realize the nonreciprocal couplings experimentally in electronic, photonic, acoustic, and mechanical systems. In this section, we validate the feasibility of realizing non-Hermitian Dirac cones in acoustic and mechanical systems with existing schemes of nonreciprocal couplings, by presenting numerically calculated dispersions using realistic parameters.

A. Acoustic implementation of non-Hermitian Dirac cones

In acoustics, we follow the scheme proposed in Ref. [4, 5] to realize the non-Hermitian Dirac cone model in this work. As schematically shown in Fig. R2a, the structure consists of identical air cavities (blue and red circles) with a fundamental dipole resonance frequency f_0 , and an intrinsic loss γ_0 . The coupling methods are shown in Fig. R2b. The reciprocal nearest-neighbor coupling t is realized by connecting air cavities with narrow tubes. The nonreciprocal coupling $\tilde{\kappa} = \rho e^{i\theta}$ with tunable amplitude ρ and phase θ is realized with a unidirectional coupler, which consists of a microphone, an amplifier, a phase shifter, and a loudspeaker. The corresponding Bloch Hamiltonian

reads

$$H^a(\mathbf{k}) = \sum_{i=0}^3 h_i^a(\mathbf{k}) \sigma_i, \quad (1)$$

where $h_0^a = \sum_{i=1,2,3} f_0 - i\gamma + \tilde{\kappa}[\cos(\mathbf{k} \cdot \mathbf{a}_i) - i \sin(\mathbf{k} \cdot \mathbf{a}_i)]$, $h_1^a = t[1 + \cos(\mathbf{k} \cdot \mathbf{a}_2) + \cos(\mathbf{k} \cdot \mathbf{a}_3)]$, $h_2^a = t[\sin(\mathbf{k} \cdot \mathbf{a}_2) - \sin(\mathbf{k} \cdot \mathbf{a}_3)]$, $h_3^a = 0$, σ_0 is the identity matrix, and $\sigma_{1,2,3}$ are the Pauli matrices. Here $\mathbf{k} = (k_x, k_y)$ is the wavevector, $\mathbf{a}_1 = (1, 0)^T$, $\mathbf{a}_2 = (-1/2, \sqrt{3}/2)^T$ and $\mathbf{a}_3 = (-1/2, -\sqrt{3}/2)^T$ are lattice vectors.

Here we use the parameters that have been realized in Ref. [5], i.e., $f_0 = 5178$ Hz, $\gamma_0 = 27.5$ Hz, $t = 44.4$ Hz, $\rho = 6.44$ Hz, and $\theta = 0$. Figure R2c shows the calculated dispersion. Two Dirac cones with different imaginary eigenfrequencies can be clearly seen at the K and K' valley, similar to the circuit case.

FIG. R2. Acoustic implementation of non-Hermitian Dirac cones. **a**, The schematic diagram of the designed acoustic structure realizing the hexagonal cell. Identical air cavities (red and blue circles) have a fundamental dipole resonance frequency f_0 and an intrinsic loss γ_0 . **b**, Coupling mechanisms. The reciprocal nearest-neighbor coupling t is realized by connecting air cavities with narrow tubes. The nonreciprocal coupling $\tilde{\kappa}$ is realized with a unidirectional coupler, which consists of a microphone, an amplifier, a phase shifter, and a loudspeaker. **c**, Calculated bulk dispersion of the acoustic lattice with parameters $f_0 = 5178$ Hz, $\gamma_0 = 27.5$ Hz, $t = 44.4$ Hz, and $\tilde{\kappa} = 6.44$ Hz. The colors denote the imaginary part of the eigenfrequencies.

B. Mechanical implementation of non-Hermitian Dirac cones

In mechanics, we follow the scheme proposed in Ref. [6, 7] to realize the non-Hermitian Dirac cones. The schematic diagram is shown in Fig. R3a. The building block is a rotational oscillator consisting of a brushless motor and a rotational arm connected to two anchor points via two tensioned springs with the on-site resonant frequency f_0 and the on-site loss γ . The coupling methods are shown in Fig. R3b. The oscillators are connected by tensioned springs to produce the recip-

rocal nearest-neighbor coupling t . The nonreciprocal coupling $t_2 \pm \delta$ is realized by programmed external actuation of the motors. The corresponding Bloch Hamiltonian reads

$$H^m(\mathbf{k}) = \sum_{i=0}^3 h_i^m(\mathbf{k}) \sigma_i, \quad (2)$$

where $h_0^b = \sum_{i=1,2,3} f_0 - i\gamma + 2[t_2 \cos(\mathbf{k} \cdot \mathbf{a}_i) - i\delta \sin(\mathbf{k} \cdot \mathbf{a}_i)]$, $h_1^m = t[1 + \cos(\mathbf{k} \cdot \mathbf{a}_2) + \cos(\mathbf{k} \cdot \mathbf{a}_3)]$, $h_2^m = t[\sin(\mathbf{k} \cdot \mathbf{a}_2) - \sin(\mathbf{k} \cdot \mathbf{a}_3)]$, $h_3^m = m$, σ_0 is the identity matrix, and $\sigma_{1,2,3}$ are the Pauli matrices. Here $\mathbf{k} = (k_x, k_y)$ is the wavevector, $\mathbf{a}_1 = (1, 0)^T$, $\mathbf{a}_2 = (-1/2, \sqrt{3}/2)^T$ and $\mathbf{a}_3 = (-1/2, -\sqrt{3}/2)^T$ are lattice vectors.

Here we use the parameters that have been realized in Ref. [7], i.e., $f_0 = 54.67$ Hz, $\gamma = 1.82$ Hz, $t = -2.33$ Hz, $t_2 = -0.38$ Hz, and $\delta = -0.1$ Hz. Figure. R3c shows the calculated dispersion. Two Dirac cones with different imaginary eigenfrequencies can be clearly seen at the K and K' valley, similar to the circuit case.

FIG. R3. Mechanical implementation of non-Hermitian Dirac cones. **a**, The schematic diagram of the designed mechanical structure realizing the hexagonal unit cell. Identical rotational oscillators (red and blue circles) have the on-site resonant frequency f_0 and an intrinsic loss γ . **b**, Coupling mechanisms. The reciprocal nearest-neighbor coupling t is realized by connecting rotational oscillators with tensioned springs. The nonreciprocal coupling $t_2 \pm \delta$ is realized by programmed external actuation of the motors. **c**, Calculated bulk dispersion of the acoustic lattice with parameters $f_0 = 54.67$ Hz, $\gamma = 1.82$ Hz, $t = -2.33$ Hz, $t_2 = -0.38$ Hz, and $\delta = -0.1$ Hz. The colors denote the imaginary part of the eigenfrequencies.

Reviewer #2 Comment 2:

I find that the non-Hermitian Dirac cones and the topological kink states are of fundamental interest. However, it is not clear to me what are the possible applications when implementing them in the electric circuit. I think that the authors need to present the motivation to realize the non-Hermitian Dirac cones in the electric circuit.

Response from Authors:

We thank the reviewer for acknowledging the fundamental interest of our work.

We choose electric circuits as the platform due to their wide variety of electrical elements and highly flexible connectivity. These features have made electric circuits a versatile and popular framework for exploring unconventional topological states [Phys. Rep. 1093, 1–54 (2024)], including non-Hermitian, nonlinear, non-Abelian, non-periodic, non-Euclidean, and higher-dimensional topological states. In our work, we have realized non-Hermitian Dirac cones in electric circuits, featuring valley-dependent lifetimes that give rise to valley kink states characterized by valley-locked lifetimes. These states enable effectively unidirectional propagation and exhibit enhanced resistance to inter-valley scattering.

In practical applications, as non-reciprocal couplings are well-controlled non-Hermitian parameters in both classical and quantum systems (as noted in response to Comment 1), the proposed non-Hermitian Dirac cones with valley-dependent lifetimes can be extended to a wide range of platforms. This opens up promising opportunities for applications, such as in lasers [Light Sci. Appl. 11, 336 (2022)] and one-way waveguides.

With regard to applications in electric circuits, our study also suggests unique ways to control electrical signal control in devices such as logic devices and switches. Additionally, topological circuits demonstrate remarkable robustness in maintaining circuit functionality despite physical deformations such as bending, folding, twisting, compressing, or stretching. This robustness positions topological circuits as a promising candidate for flexible electronics, which enable the integration of circuits on bendable substrates like polyimide or polyester films [Small Methods 2, 1800070 (2018); Adv. Mater. 19, 1897–1916 (2007)]. Furthermore, topological circuits can be fully realized using complementary metal-oxide-semiconductor (CMOS) technology [Nat. Nanotechnol. 17, 262–268 (2022); Nat. Electron. 5, 300–309 (2022)], which may solve the critical challenges met in the chip industry.

To make the reader aware of our motivation and the potential applications in circuits, we have made the following revisions in the main text:

On page 3, right column: “We experimentally study this tight-binding model using an electric circuit lattice, which has been a popular platform for studying complex physical models **because of the wide variety of electrical elements and highly flexible connectivity it offers** [24-30]”.

On page 7, left column: “At the boundary, topological valley kink states are found to inherit the non-Hermitian physics in the bulk and exhibit unidirectional propagation and suppressed inter-

valley scattering that are not seen in the Hermitian case, **which could be used for robust logic gates and compact circuit integration**".

Reviewer #2 Comment 3:

In addition to the above two points, I also would like the authors to consider the following questions and comments.

In the caption of Fig. 1, the authors have stated "The bandwidth for the single Dirac cone is determined by the gap size of the K' valley, ..." regarding Fig. 1b, but the meaning of "the bandwidth of the single Dirac cone" is not clearly stated. Naively, the bandwidth of the Dirac cone is determined by to what extent the linear approximation of the Hamiltonian works near the Dirac point and has nothing to do with the information of the opposite valley. I suppose that the authors mean as follows: Within the energy range in the band gap of K' valley, the only existing eigenmodes are the massless Dirac cone at K point. So I would like the authors to clarify the meaning of the bandwidth of the single Dirac cone in this context.

Response from Authors:

We thank the reviewer for raising this question. We have modified the corresponding sentences in the main text:

Page 2, caption of Fig. 1: For the Haldane model, the TRS breaking leads to a single Dirac cone in the K valley **in the energy range within the bandgap of the K' valley, as denoted by the orange dashed lines.**

Page 2, caption of Fig. 1: **In the long time limit, only states at the K valley will survive due to the sharp contrast in lifetime between the two valleys, leading to an effective single Dirac cone behavior over a large energy range as indicated by the orange dashed lines.**

Page 3, left column: **Moreover, the energy range for having states in a single valley is much larger in our case as the imaginary eigenvalue contrasting goes far beyond the low-energy limit (see Fig. 1c).**

Besides, **the text "single Dirac cone" has been moved from the lower panel of Fig. 1c.**

Reviewer #2 Comment 4:

In the electric circuits, it is possible to realize the periodic boundary condition by connecting the nodes at both ends (in each direction) directly, while the actual circuit in Fig. 2a seems to correspond to the open boundary condition. I think assigning the periodic boundary condition is advantageous in that the obtained spectrum can be directly compared to the band structure in

the momentum-space picture. I wonder why the authors implement the open boundary condition rather than the periodic boundary condition.

Response from Authors:

Indeed, periodic boundary conditions (PBCs) can be implemented in electric circuits. In our work, we choose to implement open boundary conditions (OBCs) with absorbing resistors for the following reasons:

1. Experimental stability: Under PBCs, the amplifying modes will keep growing their intensities and thus make the circuit unstable. After introducing absorbing resistors at the boundaries (i.e., OBCs with absorbing resistors), the reflections are eliminated, and the circuits will be more stable. The absorbing boundaries also minimize the non-Hermitian skin effect and make the measured dispersion consistent to the PBC dispersion.

2. Experimental convenience: In our work, four configurations are experimentally demonstrated, including massless bulk, massive bulk, zigzag interface, and armchair interface. In practice, each configuration is constructed with small pieces of printed circuit board (PCB) (e.g., see Fig. 2a in the main text). By setting all configurations with OBC, the types of small PCB pieces are the same for each configuration, facilitating the PCB design and improving the reusability of PCB pieces.

Reviewer #2 Comment 5:

In Figs. 2c, 3a, 4a, 5b and 5c, the authors plotted the “calculated dispersion”, where the vertical axis is $Re(f)$, but it is not clear to me how this quantity is related to the eigenvalue equation of Eq. (3). It seems to me that the plotted curves correspond to the real part of the eigenvalues of $H^c(k)$ [i.e., $E(\omega)$], but the eigenvalue is not the frequency (i.e., ω) itself, as the authors state in the same paragraph as Eq. (3). Thus, I would like to authors to clarify what is the definition of $Re(f)$ and how it is related to $E(\omega)$ or ω in Eq. (3).

Response from Authors:

As indicated by Eq. (3) in the manuscript, there is a one-to-one correspondence between the eigenvalues in the tight-binding model and the eigenfrequencies in the circuit model: $E(\omega) = 1/\omega^2 L - i/\omega R_s$. The eigenvalues $E(\omega)$ are initially obtained by numerically diagonalizing the Hamiltonian, after which the above equation is solved to retrieve the corresponding eigenfrequencies $f = \omega/2\pi$. And $Re(f)$ denotes the real part of the eigenfrequency.

Following the reviewer’s suggestion, we have added one sentence on page 4, right column, to clarify this issue, which reads “**The circuit dispersion (i.e., the relationship between frequency and**

momentum) can be obtained by first solving Eq. (3) and then mapping the eigenvalue $E(\omega)$ to the frequency regime through $E(\omega) = 1/\omega^2 L - i/\omega R_s$.

Reviewer #2 Comment 6:

In the right column of page 6, the authors have stated "... thus restore gapless transport". I think the word "gapless transport" is ambiguous, so the authors will need to clarify what it means. In the electric circuit, the kink states give rise to characteristic voltage distribution but their relation to the transport phenomena is unclear.

Response from Authors:

We are grateful to the reviewer for bringing this issue to our attention. The term "transport" has been modified to "dispersion".

Reviewer #3

General comments from Reviewer #3:

The paper by Xinrong Xie et al. is an experimental work on the realization of non-Hermitian Dirac cones in circuits. By introducing non-reciprocal next-nearest-neighbor couplings via circuit elements, the authors first experimentally demonstrate the massless and massive non-Hermitian Dirac cones at K and K' points. Afterward, topological valley kink states of a zigzag interface and parity-time transition of kink states at an armchair interface have been carefully investigated. The results are clear, and therefore, I would suggest its publication in Nature Communications if the authors can address the following issues:

Response from Authors:

We thank the reviewer for carefully reading our manuscript and judging that *the investigation is careful and the results are clear*. We sincerely thank the reviewer for the very thoughtful and constructive comments. We address them point-by-point as follows.

Specific comments from Reviewer #3:

Reviewer #3 Comment 1:

It is known that when a non-Hermitian system has skin modes, all the properties, including topological invariants, band dispersions, and so on, shall be investigated on the generalized Brillouin zone (BZ) but not on the BZ. From my understanding (for example, see Fig. S6), the system under consideration does not have skin modes and thus can focus on the BZ, which is the current approach used throughout the manuscript. However, can the authors clearly prove and demonstrate this issue in the main text?

Response from Authors:

Thank you for your insightful observation. Indeed, for non-Hermitian systems exhibiting skin modes, the generalized Brillouin zone (BZ) is crucial to resolve the mismatch between bulk properties under periodic boundary conditions (PBCs) and boundary phenomena under open boundary conditions (OBCs). However, in our study, we have introduced absorbing resistors at the system's boundaries. This approach effectively eliminates skin modes by eliminating the reflection at the boundaries, ensuring that the conventional BZ remains a valid framework for analyzing the system. This can already be seen by our measured dispersions on finite-size samples (e.g., Fig. 2d and Fig. 4c), which agree well with PBC dispersions.

To better clarify this point, we have added the one sentence on page 4, left column: “**With the**

absorbing resistors, the non-Hermitian skin effect induced by the boundaries is suppressed and we will work on the conventional Brillouin zone throughout this work”.

Reviewer #3 Comment 2:

All the experimental results, especially the bulk ones, can hardly tell the dispersions solely. Does such an issue come from sample sizes, intrinsic losses, or some other systematic reasons? Please discuss this issue and possible improvements in the main text.

Response from Authors:

We thank the reviewer for pointing out this issue, which arises due to the sharp contrast between the Fourier intensity.

The Fourier intensity is proportional to the amplitude of the (steady-state) voltage field of the excited modes, which is influenced by the imaginary part of the eigenfrequencies, i.e., a larger imaginary part leads to a larger mode amplitude. Thus, for a non-Hermitian system, the measured dispersion from the Fourier transform of a steady-state field distribution would only match part of the calculated dispersion where the modes have larger lifetimes.

In the case of the massive bulk structure, some eigenmodes exhibit a significantly larger imaginary part, as indicated by the purple circle in the upper panel of Fig. R4a, rendering them dominant in the measured dispersion (Fig. R4a, lower panel). In contrast, the massless case, as well as the zigzag and armchair interfaces (Fig. R4b-d, upper panel), exhibit a more uniform distribution of imaginary parts of eigenfrequencies, leading to more evenly measured dispersions (Fig. R4b-d, lower panel). The proportional relation between the Fourier intensity and the imaginary part of the eigenfrequencies is also observed, as evidenced by the consistency between the purple circles and the relatively large Fourier intensities in the measured dispersions (Fig. R4b-d, lower panel).

We have added one sentence to explain this issue on page 5, left and right columns: We excite the circuit at sublattices A and B, respectively, to measure the upper and lower bands, as given in Fig. 3b-d. It is observed that wherever the circuit is excited at sublattice A or B, only the dispersion at the K valley can be seen, similar to the massless case. **Due to the significant contrast between imaginary parts of eigenfrequencies, modes with much larger imaginary parts (i.e., those near the band edges) dominate the measured dispersion.** Note that the upper band is also excited when we excite at sublattice B since the upper band has a larger imaginary part than the lower band (see Fig. 3a).

FIG. R4. Comparison between measured results of different schemes. Calculated (upper panel) and measured (lower panel) dispersions of massive (a), massless (b), zigzag interface (c), and armchair interface (d). Purple circles denote the eigenmodes with relatively large imaginary eigenfrequencies in the calculated dispersion.

Reviewer #3 Comment 3:

Figure S6(a): Can the authors clearly identify which regions inside the real energy line gap have non-zero winding numbers (or, equivalently, point gaps nontrivially open), especially the region near $Re(E)=0.2$?

Response from Authors:

We have added the calculation results of the winding number of point gaps and revised Figure S6(a) for clearer demonstration. The corresponding revisions are as follows.

Supplementary Information, Page 6: From the non-Hermitian topology point of view, the non-Hermitian valley interface mimics the classic Hatano-Nelson model with a non-trivial point-gap topology [2]. When periodic boundary conditions are imposed, the non-Hermitian zigzag valley interface exhibits a set of delocalized states with pair-wise complex eigenvalues, forming a point gap in the spectrum as illustrated by black dots in Fig. R5a. **The topological properties of the point gap are characterized by the winding number [3]:**

$$w = -i \int_{\text{BZ}} \frac{dk}{2\pi} \text{Tr}[Q(k)], \quad (3)$$

with $Q(k) = [H(k) - E]^{-1} \partial_k [H(k) - E]$, E is any complex value in the point gap, and $\text{BZ} = [-\pi, \pi]$ represents the Brillouin zone. By choosing $E = 0.2$, we find that the point gap is topologically non-trivial, characterized by a nonzero winding number of $w = 1$.

Beside, the demonstration of localized eigenstates in Fig. R5b has been updated to focus the eigenstate corresponding to an eigenvalue near $E = 0.2$.

FIG. R5. Point-gap topology of the valley-kink states **a**, Eigenvalues of the non-Hermitian zigzag valley interface when periodic (black dots), or open (blue dots) boundary conditions are applied. The nearest-neighbor coupling coefficient is $t = 1$, the NNN coupling coefficient is $t_2 \pm \delta = 0.1 \pm 0.1$, and the mass term is $m = \pm 0.5$. **b**, The eigenstate distribution corresponding to the eigenvalue highlighted with a black arrow in (a). The green dashed line denotes the zigzag interface.

Reviewer #3 Comment 4:

If the authors perform similar measurements in Figs. 5(d-f) on Fig. 5(b), how about the experimental results?

Response from Authors:

We thank the reviewer for raising this question.

To address this question, we would like to present the simulated gapped dispersion of the non-Hermitian armchair model using the LTSpice software. Prior to this, the accuracy of the simulation is validated by comparing the simulated (Fig. R6a) and measured (Fig. R6b) dispersions of gapless armchair interfaces. The circuit parameters of the simulation models are identical to those of the experimental models. As illustrated in Fig. R6, the simulated and measured dispersions agree well with each other, thereby validating the accuracy of the simulation results.

Fig. R7a shows the calculated dispersion of the non-Hermitian armchair interface with $C_2/C_1 = 0.1$, which reveals that the edge states exhibit smaller contrasting imaginary eigenfrequencies than the model with $C_2/C_1 = 1$. This leads to the formation of a band gap at $k_y = 0$.

We then employ LTSpice software to conduct the simulation, where the model has the identical size as the experimental setup described in the main text. The simulated (colormaps) and calculated (dashed lines) projected dispersion for the armchair valley kink states are shown in Fig. R7b

and Fig. R7c, corresponding to the excitations at the bottom and top, respectively. The results validate the small contrast of the imaginary eigenfrequencies in the calculated results, where we observe the existence of the band gap and the dispersion of edge modes that propagate downwards.

FIG. R6. Comparison between simulated and measured dispersions of gapless armchair interfaces. Simulated (a) and measured (b) edge dispersion from Fourier transform of the fields at the interface with excitations at the bottom (upper panel) and top (lower panel) sides, respectively. The armchair interface is constructed with a lattice with $C_a = 100$ nF and $C_b = 200$ nF at the left domain and a lattice with $C_a = 200$ nF and $C_b = 100$ nF at the right domain. Other parameters are $C = 100$ nF, $C_1 = 10$ nF, $C_2 = 10$ nF, $L = 1.043$ μ H, $R_s = 30$ Ω . The colors represent the normalized Fourier intensity and the dashed lines represent the theoretical results.

FIG. R7. Simulated dispersion of gapped armchair interfaces. **a**, Calculated dispersion of a heterostructure with an armchair interface constructed with a lattice with $C_a = 100$ nF and $C_b = 200$ nF at the left domain and a lattice with $C_a = 200$ nF and $C_b = 100$ nF at the right domain. Other parameters are $C = 100$ nF, $C_1 = 10$ nF, $C_2 = 1$ nF, $L = 1.043$ μ H, $R_s = 1$ M Ω . The colors denote the imaginary part of eigenfrequencies. **b-c**, Simulated edge dispersion from Fourier transform of the fields at the interface with excitations at the bottom (**b**) and top (**c**) sides, respectively. The colors represent the normalized Fourier intensity and the dashed lines represent the theoretical results.

Response Letter to Reviewers

We are grateful for the constructive comments on this manuscript (NCOMMS-24-62468A) from all reviewers. In the text below, the comments from each reviewer are quoted in *blue italics*, and followed by our response.

Reviewer #1

General comments from Reviewer #1:

I thank the authors for their comprehensive response to the comments on their manuscript. I find their revisions sufficient to address my remarks, so I recommend it to be published in Nature Communications.

Response from Authors:

We thank the reviewer for the recommendation for the publication of this work.

Reviewer #2

General comments from Reviewer #2:

The authors have considered the questions and comments I raised in the previous report and they have revised the manuscript and the supplementary information. They have provided a convincing explanation that their finding of the non-Hermitian Dirac cones with the valley-dependent life time will have broad potential applications not only for the electric circuits but also for various other physical setups. Therefore, I recommend the publication of this manuscript in Nature Communications.

Response from Authors:

We thank the reviewer for the recommendation for the publication of this work.

Reviewer #3

General comments from Reviewer #3:

The authors have addressed my concerns, so I suggest its publication in its current form now.

Response from Authors:

We thank the reviewer for the recommendation for the publication of this work.